# Meta Architecture Search

**Albert Shaw**[1]*    **Wei Wei**[2]    **Weiyang Liu**[1]    **Le Song**[1,3]    **Bo Dai**[1,2]
[1]Georgia Institute of Technology    [2]Google Research    [3]Ant Financial

## Abstract

Neural Architecture Search (NAS) has been quite successful in constructing state-of-the-art models on a variety of tasks. Unfortunately, the computational cost can make it difficult to scale. In this paper, we make the first attempt to study Meta Architecture Search which aims at learning a task-agnostic representation that can be used to speed up the process of architecture search on a large number of tasks. We propose the Bayesian Meta Architecture SEarch (BASE) framework which takes advantage of a Bayesian formulation of the architecture search problem to learn over an entire set of tasks simultaneously. We show that on `Imagenet` classification, we can find a model that achieves 25.7% top-1 error and 8.1% top-5 error by adapting the architecture in less than an hour from an 8 GPU days pretrained meta-network. By learning a good prior for NAS, our method dramatically decreases the required computation cost while achieving comparable performance to current state-of-the-art methods - even finding competitive models for unseen datasets with very quick adaptation. We believe our framework will open up new possibilities for efficient and massively scalable architecture search research across multiple tasks.

## 1 Introduction

For deep neural networks, the particular structure often plays a vital role in achieving state-of-the-art performance in many practical applications, and there has been much work [16, 11, 13, 41, 23, 22, 21, 32, 31, 36] exploring the space of neural network designs. Due to the combinatorial nature of the design space, hand-designing architectures is time-consuming and inevitably sub-optimal. Automated Neural Architecture Search (NAS) has had great success in finding high-performance architectures. However, people may need optimal architectures for several similar tasks at once, such as solving different classification tasks or even optimizing task networks for both high accuracy and efficient inference on multiple hardware platforms [35]. Although there has been success in transferring architectures across tasks [43], recent work has increasingly shown that the optimal architectures can vary between even similar tasks; to achieve the best results, NAS would need to be repeatedly run for each task [5] which can be quite costly.

In this work, we present a first effort towards Meta Architecture Search, which aims at learning a task-agnostic representation that can be used to search over multiple tasks efficiently. The overall graphical illustration of the model can be found in Figure 1, where the meta-network represents the collective knowledge of architecture search across tasks. Meta Architecture Search takes advantage of the similarities among tasks and the corresponding similarities in their optimal networks, reducing the overall training time significantly and allowing fast adaptation to new tasks. We formulate the Meta Architecture Search problem from a Bayesian perspective and propose Bayesian Meta Architecture SEarch (BASE), a novel framework to derive a variational inference method to learn optimal weights and architectures for a task distribution. To parameterize the architecture search space, we use a stochastic neural network which contains all the possible architectures within our architecture

The code repository is available at https://github.com/ashaw596/meta_architecture_search.

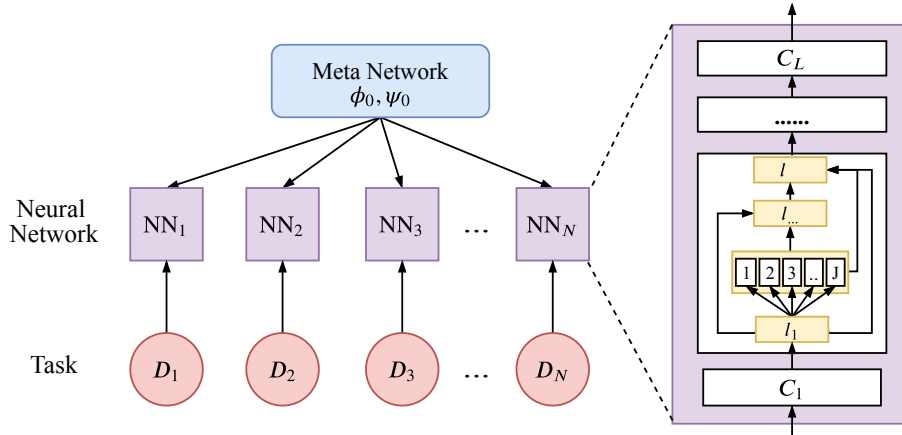

Figure 1: Illustrations of Meta Architecture Search. We train a shared distribution for the meta-network and a sample from the distribution will quick adapt to new task.

space as specific paths within the network. By using the Gumbel-softmax [14] distribution in the parameterization of the path distributions, this network containing an entire architecture space can be optimized differentially. To account for the task distribution in the posterior distribution of the neural network architecture and weights, we exploit the optimization embedding[6] technique to design the parameterization of the posterior. This allows us to train it as a meta-network optimized over a task distribution.

To train our meta-network over a wide distribution of tasks with different image sizes, we define a new space of classification tasks by randomly selecting 10 `Imagenet` [7] classes and downsampling the images to $32 \times 32$, $64 \times 64$, or $224 \times 224$ image sizes. By training on these datasets, we can learn good distributions of architectures optimized for different image sizes. With a meta-network trained for 8 GPU days, we then show that we can achieve very competitive results on full `Imagenet` by deriving optimal task-specific architectures from the meta-network, obtaining 25.7% top-1 error on `ImageNet` using an adaption time of less than one hour. Our method achieves significantly lower computational costs compared to current state-of-the-art NAS approaches. By adapting the multi-task meta-network for to the unseen `CIFAR10` dataset for less than one hour, we found a model that achieves 2.83% Top-1 Error. Additionally, we also apply this method to tackle neural architecture search for few-shot learning, demonstrating the flexibility of our framework.

Our research opens new potentials for using Meta Architecture Search across massive amounts of tasks. The nature of the Bayesian formulation makes it possible to learn over an entire collection of tasks simultaneously, bringing additional benefits such as computational efficiency and privacy when performing neural architecture search.

## 2 Related Work

**Neural Architecture Search** Several evolutionary and reinforcement learning based algorithms have been quite successful in achieving state-of-the-art performances on many tasks [42, 43, 30, 12]. However, these methods are computationally costly and require tremendous amounts of computing resources. While previous work has achieved good results with sharing architectures across tasks [43], [35] and [5] show that task and even platform-specific architecture search is required in order to achieve the best performance. Several methods [20, 27, 4, 3, 17] have been proposed to reduce the search time, and both FBNet [35] and SNAS [37] utilize the Gumbel-Softmax [14] distribution similarly to our meta-network design to allow gradient-based architecture optimization. [2] and [40] also both propose methods to generate optimal weights for one task given any architecture like our meta-network is capable of. Their methods, however, do not allow optimization of the architectures and are only trained on a single task making them inefficient in optimizing over multiple tasks. Similarly to our work, [34] recently proposed methods to accelerate search utilizing knowledge from previous searches and predicting posterior distributions of the optimal architecture. Our approach, however, achieves much better computational efficiency by not limiting ourselves to transferring knowledge from only the performance of discrete architectures on the validation datasets, but instead

sharing knowledge for both optimal weights and architecture parameters and implicitly characterizing the entire dataset utilizing optimization embedding.

**Meta Learning**   Meta-learning methods allow networks to be quickly trained on new data and new tasks [8, 29]. While previous works have not applied these methods to Neural Architecture Search, our derived Bayesian optimization method bears some similarities to Neural Processes [9, 10, 15]. Both can derive a neural network specialized for a dataset by conditioning the model on some samples from the dataset. The use of neural networks allows both to be optimized by gradient descent. However, Neural Processes use specially structured encoder and aggregator networks to build a context embedding from the samples. We use the optimization embedding technique [6] to condition our neural network using gradient descent in an inner loop, which allows us to avoid explicitly summarizing the datasets with a separate network. This inner-outer loop dynamic shares some similarities to second-order MAML [8]. Both algorithms unroll the stochastic gradient descent step. Due to this, we are also able to establish a connection between the heuristic MAML algorithm and Bayesian inference.

## 3   A Bayesian Inference View of Architecture Search

In this section, we propose a Bayesian inference view for neural architecture search which naturally introduces the hierarchical structures across different tasks. Such a view inspires an efficient algorithm which can provide a task-specific neural network with adapted weights and architecture using only *a few* learning steps.

We first formulate the neural architecture search as an operation selection problem. Specifically, we consider the neural network as a composition of $L$ layers of cells, where the cells share the same architecture, but have different parameters. In the $l$-th layer, the cell consists of a $K$-layer sub-network with bypass connections. Specifically, we denote the $x_k^l$ as the output of the $k$-th layer of $l$-th cell

$$x_k^l = \sum_{i=1}^{k-1} \left( z_{i,k}^\top \mathcal{A}_i \left( \theta_{i,k}^l \right) \right) \circ x_i^l := \sum_{i=1}^{k-1} \sum_{j=1}^{J} z_{ij,k} \phi_i^j \left( x_i^l; \theta_{ij,k}^l \right) \tag{1}$$

where $\mathcal{A}_i \left( \theta_{i,k}^l \right) = \left[ \phi_i^j \left( \cdot; \theta_{ij,k}^l \right) \right]_{j=1}^{J}$ denotes a group of $J$ different operations from $\mathbb{R}^d \to \mathbb{R}^p$ which depend on parameters $\theta_{ij,k}^l$, *e.g.*, different nonlinear neurons, convolution kernels with different sizes, or other architecture choices. $z_{i,k}$ are all binary variables which are shared across $L$ layers. They indicate which layers from the 1 to $k-1$ levels in $l$-th cell should be selected as inputs to the $k$-th layer. Therefore, with different instantiations of $z$, the cell will select different operations to form the output. Figure 1 has an illustration of this structure.

We assume the probabilistic model as

$$\begin{aligned} \theta_k^l := \left[ \theta_{ij,k}^l \right]_{i,j=1}^{k-1,J} &\sim \mathcal{N} \left( \mu_k^l, \left( \sigma_k^l \right)^2 \right), \\ z_{i,k} &\sim \mathcal{C}ategorial \left( \alpha_{i,k} \right), \; k = 1, \dots, K, \\ y &\sim p \left( y | x; \theta, z \right) \propto \exp \left( -\ell \left( f \left( x; \theta, z \right), y \right) \right), \end{aligned} \tag{2}$$

with $\theta = \left\{ [\theta_k^l]_{l=1}^L \right\}_{k=1}^K$, $z = \left\{ [z_{i,k}]_{i=1}^{k-1} \right\}_{k=1}^K$, and $\alpha_{i,k}^l \geqslant 0$, $\sum_{l=1}^L \alpha_{i,k}^l = 1$. With this probabilistic model, the selection of $z$, *i.e.*, neural network architecture search, is reduced to finding a distribution defined by $\alpha$, and the neural network learning is reduced to finding $\theta$, both of which are the parameters of the probabilistic model.

The most natural choice here for probabilistic model estimation is the maximum log-likelihood estimation (MLE), *i.e.*,

$$\max_{W:=(\mu,\sigma,\alpha)} \widehat{\mathbb{E}}_{x,y} \left[ \log \int p \left( y | x; \theta, z \right) p \left( z; \alpha \right) p \left( \theta; \mu, \sigma \right) dz d\theta \right]. \tag{3}$$

However, the MLE is intractable due to the integral over latent variable $z$. We apply the classic variational Bayesian inference trick, which leads to the evidence lower bound (ELBO), *i.e.*,

$$\max_W \max_{q(z), q(\theta)} \; -\widehat{\mathbb{E}}_{x,y} \mathbb{E}_{z \sim q(z), \theta \sim q(\theta)} [\ell \left( f \left( x; \theta, z \right), y \right)] - KL \left( q(z) q(\theta) || p \left( z, \theta \right) \right), \tag{4}$$

where $p \left( z \right) = \prod_{k=1}^K \prod_{i=1}^{k-1} \mathcal{C}ategorial \left( z_{i,k} \right) = \prod_{k=1}^K \prod_{i=1}^{k-1} \prod_{l=1}^L \left( \alpha_{i,k}^l \right)^{z_{i,k}^l}$. As shown in [39], the optimal solution of (4) in all possible distributions will be the posterior. With such a model, architecture learning can be recast as Bayesian inference.

### 3.1 Bayesian Meta Architecture Learning

Based on the Bayesian view of architecture search, we can easily extend it to the meta-learning setting, where we have many tasks, *i.e.*, $\mathcal{D}_t = \{x_i^t, y_i^t\}_{i=1}^n$. We are required to learn the neural network architectures and the corresponding parameters jointly while taking the task dependencies on the neural network structure into account.

We generalize the model (2) to handle multiple tasks as follows. For the $t$-th task, we design the model following (2). Meanwhile, the hyperparameters, *i.e.*, $(\mu, \sigma, \alpha)$, are shared across all the tasks. In other words, the layers and architecture priors are shared between tasks. Then we have the MLE:

$$\max_W \ \widehat{\mathbb{E}}_{\mathcal{D}_t} \widehat{\mathbb{E}}_{(x,y) \sim \mathcal{D}_t} \left[ \log \int p\left(y|x; \theta, z\right) p\left(z; \alpha\right) p\left(\theta; \mu, \sigma\right) dz d\theta \right] \tag{5}$$

Similarly, we exploit the ELBO. Due to the structures induced by sharing across the tasks, the posteriors for $(z, \theta)$ have special dependencies, *i.e.*,

$$\max_W \widehat{\mathbb{E}}_{\mathcal{D}_t} \left( \max_{q(z|\mathcal{D}), q(\theta|\mathcal{D})} \widehat{\mathbb{E}}_{(x,y) \sim \mathcal{D}_t} \mathbb{E}_{z \sim q(z|\mathcal{D}), \theta \sim q(\theta|\mathcal{D})} \left[ -\ell\left(f\left(x; \theta, z\right), y\right) \right] - KL\left(q||p\right) \right) \tag{6}$$

With the variational posterior distributions, $q\left(z|\mathcal{D}\right)$ and $q\left(\theta|\mathcal{D}\right)$, introduced into the model, we can directly generate the architecture and its corresponding weights based on the posterior. In a sense, the posterior can be understood as the neural network predictive model.

## 4 Variational Inference by Optimization Embedding

The design of the parameterization of the posterior $q\left(z|\mathcal{D}\right)$ and $q\left(\theta|\mathcal{D}\right)$ is extremely important, especially in our case where we need to model the dependence between $(z, \theta)$ w.r.t. the *task distributions* $\mathcal{D}$ and the *loss information*. Fortunately, we can bypass this problem by applying parameterized Coupled Variational Bayes (CVB), which generates the parameterization automatically through *optimization embedding* [6].

Specifically, we assume the $q\left(\theta|\mathcal{D}\right)$ is Gaussian and the $q\left(z|\mathcal{D}\right)$ is a product of the categorical distribution. We approximate the categorical $z$ with the Gumbel-Softmax distribution [14, 25], which leads to a valid gradient so that the model will be fully differentiable. Therefore, we have

$$q_\psi\left(\theta|\mathcal{D}\right) = \mathcal{N}\left(\psi_\mu, \psi_\sigma\right), \quad q_\phi\left(z_{i,k}|\mathcal{D}\right) = \Gamma\left(r\right) \tau^{L-1} \left( \sum_{l=1}^L \frac{\pi_{\mathcal{D}, \phi_{i,k}^l, l}}{\left(z_{i,k}^l\right)^\tau} \right)^{-r} \prod_{i=1}^r \left( \frac{\pi_{\mathcal{D}, \phi_{i,k}^l, l}}{\left(z_{i,k}^l\right)^{\tau+1}} \right) \tag{7}$$

Then, we can sample $(\theta, z)$ by following,

$$\theta_\mathcal{D}\left(\epsilon, \psi\right) = \psi_{\mathcal{D}, \mu} + \epsilon \psi_{\mathcal{D}, \sigma}, \quad \epsilon \sim \mathcal{N}\left(0, 1\right),$$

$$z_{i,k,\mathcal{D}}^l\left(\xi, \phi\right) = \frac{\exp\left(\left(\phi_{\mathcal{D},i,k}^l + \xi^l\right)/\tau\right)}{\sum_{l=1}^L \exp\left(\left(\phi_{i,k}^l + \xi^l\right)/\tau\right)}, \quad \xi^l \sim \mathcal{G}\left(0, 1\right), \quad l \in \{1, \ldots, L\}, \tag{8}$$

with $\pi_{x, \phi, i} = \frac{\exp\left(\phi_{x,i}\right)}{\sum_{i=1}^P \exp\left(\phi_{x,i}\right)}$ and $\mathcal{G}\left(0, 1\right)$ denotes the Gumbel distribution. We emphasize that we do not have any explicit form of the parameters $\phi_\mathcal{D}$ and $\psi_\mathcal{D}$ yet, which will be derived by optimization embedding automatically.

Plugging the formulation into the ELBO (6), we arrive at the objective

$$\widehat{\mathbb{E}}_\mathcal{D} \left[ \max_{\phi_\mathcal{D}, \psi_\mathcal{D}} \underbrace{\widehat{\mathbb{E}}_{x,y} \mathbb{E}_{\xi,\epsilon} \left[ -\ell\left(f\left(x; \theta_\mathcal{D}\left(\epsilon, \psi\right), z_\mathcal{D}\left(\xi, \phi\right)\right), y\right) \right] - \log \frac{q_\phi\left(z|\mathcal{D}\right)}{p\left(z; \alpha\right)} - \log \frac{q_\psi\left(\theta|\mathcal{D}\right)}{p\left(\theta; \mu, \sigma\right)}}_{L(\phi_\mathcal{D}, \psi_\mathcal{D}; W)} \right]. \tag{9}$$

With the ultimate objective (9) we follow the parameterized CVB derivation [6] for embedding the optimization procedure for $(\phi, \psi)$. Denoting the $\widehat{g}_{\phi_\mathcal{D}, \psi_\mathcal{D}}\left(\mathcal{D}, W\right) = \frac{\partial \widehat{L}}{\partial(\phi_\mathcal{D}, \psi_\mathcal{D})}$ where $\widehat{L}$ is the stochastic approximation for $L\left(\phi_\mathcal{D}, \psi_\mathcal{D}; W\right)$, then, the stochastic gradient descent (SGD) iteratively updates as

$$\left[\phi_\mathcal{D}^t, \psi_\mathcal{D}^t\right] = \eta_t \widehat{g}_{\phi_\mathcal{D}, \psi_\mathcal{D}}\left(\mathcal{D}, W\right) + \left[\phi_\mathcal{D}^{t-1}, \psi_\mathcal{D}^{t-1}\right], \tag{10}$$

We can initialize $(\phi^0, \psi^0) = W$ which is shared across all the tasks. Alternative choices are also possible, *e.g.*, with one more neural network, $(\phi^0, \psi^0) = h_V(\mathcal{D})$. We unfold $T$ steps of the iteration to form a neural network with output $(\phi_\mathcal{D}^T, \psi_\mathcal{D}^T)$. Plugging the obtained $(\phi_\mathcal{D}^T, \psi_\mathcal{D}^T)$ to (8), we have the parameters and architecture as $(\theta_\mathcal{D}^T(\xi, \psi_\mathcal{D}^T), z_\mathcal{D}(\xi, \phi_\mathcal{D}^T))$. In other words, we derive the concrete parameterization of $q(\theta|\mathcal{D})$ and $q(z|\mathcal{D})$ automatically by unfolding the optimization steps. Replacing the parameterization of $q(z|\mathcal{D})$ and $q(\theta|\mathcal{D})$ into $L(\phi_\mathcal{D}, \psi_D, W)$, we have

$$\max_W \; \widehat{\mathbb{E}}_\mathcal{D} \widehat{\mathbb{E}}_{x,y} \mathbb{E}_{\xi,\epsilon} \underbrace{\left[ -\ell\left(f\left(x; \theta_\mathcal{D}^T(\epsilon, \psi), z_\mathcal{D}^T(\xi, \phi)\right), y\right) - \log \frac{q_{\phi_\mathcal{D}^T}(z|\mathcal{D})}{p(z;\alpha)} - \log \frac{q_{\psi_\mathcal{D}^T}(\theta|\mathcal{D})}{p(\theta;\mu,\sigma)} \right]}_{\widehat{L}(x,y,\epsilon,\xi;W)}. \quad (11)$$

If we apply stochastic gradient ascent in the optimization (11) for updating $W$, the instantiated algorithm from optimization embedding shares some similarities to second-order MAML [8] and DARTS [20] algorithms. Both of these two algorithms unroll the stochastic gradient step. However, with the introduction of the Bayesian view, we can exploit the rich literature for the approximation of the distributions on discrete variables. More importantly, we can easily share both the architecture and weights

---

**Algorithm 1** Bayesian meta Architecture SEarch (BASE)

1: Initialize meta-network parameters $W_0$.
2: **for** $e = 1, \ldots, E$ **do**
3:      Sample $C$ tasks $\{\mathcal{D}_c\}_{c=1}^C \sim \mathcal{D}$.
4:      **for** $\mathcal{D}_c$ in $\mathcal{D}$ **do**
5:          Sample $\{x_t, y_t\}_{t=1}^T \sim \mathcal{D}_c$.
6:          Let $\phi_c^0, \psi_c^0 = W_{e-1}$.
7:          **for** $t = 1, \ldots, T$ **do**
8:              Sample $\xi \sim \mathcal{G}(0,1)$.
9:              Update $[\phi_c^t, \psi_c^t] = [\phi_c^{t-1}, \psi_c^{t-1}] - $
                 $\eta \nabla_{\phi_c^{t-1}, \psi_c^{t-1}} \widehat{L}(f(x_t; \phi_c^{t-1}, \psi_c^{t-1}, \xi), y_t)$.
10:      Update $W_e = W_{e-1} + \lambda \frac{1}{C} \sum_{c=1}^C ([\phi_c^T, \psi_c^T] - W_{e-1})$.

---

across many tasks. Finally, this establishes the connection between the heuristic MAML algorithm to Bayesian inference, which can be of independent interest.

**Practical algorithm:** In the method derivation, for the simplicity of exposition, we assumed there is only one cell shared across all the layers in every task, which may be overly restrictive. Following [43], we design two types of cells, named as a normal cell with $\phi_n$ and a reduction cell with $\phi_r$, which appear alternatively in the neural network. Please refer to Appendix B.3 for an illustration.

In practice, the multistep-unrolling of the gradient computation is expensive and memory inefficient. We can exploit the finite difference approximation for the gradient. This is similar to the iMAML [28] and REPTILE [26] approximations of MAML. Moreover, we can further accelerate learning by exploiting parallel computation. Specifically, for each task, we start from a local copy of the current $W$ and apply stochastic gradient ascent based on the task-specific samples. Then, the shared $W$ can be updated by summarizing the task-specific parameters and architecture. The pseudo-code for the concrete algorithm for Bayesian meta-Architecture SEarch (BASE) can be found in Algorithm 1.

With a meta-network trained with BASE over a series of tasks, for a new task, we can adapt an architecture by sampling from the posterior distribution of $z_D$ through (7) with $[\phi_D^T, \psi_D^T]$ calculated by (10) given new task $D$ which will be used to define the full-sized network. Illustrations of the network motifs used for the search network and the full networks can be found in Appendix A.2. More details about the architecture space can be found in Appendix A.

## 5 Experiments and Results

### 5.1 Experiment Setups

**Downsampled Multi-task Datasets** To help the meta-network generalize to inputs with different sizes, we create three new multi-task datasets: `Imagenet32`(Imagenet downsampled to 32x32), `Imagenet64`(Imagenet downsampled to 64x64), and `Imagenet224`(Imagenet downsampled to 224x224). `Imagenet224` uses the most commonly used size for inference for the full `Imagenet` dataset in the mobile setting. Our tasks are defined by sampling 10 random classes from one of the resized `Imagenet` datasets similar to the `Mini-Imagenet` dataset [33] in few-shot learning. This allows us to sample tasks from a space of $C(1000, 10) \times 3 \approx 2.634 \times 10^{23}$ tasks.

Table 1: Classification Accuracies on CIFAR10

| Architecture | Top-1 Test Error | Parameters (M) | Search Time (GPU Days) |
|---|---|---|---|
| NASNet-A + cutout [43] | 2.65 | 3.3 | 1800 |
| AmoebaNet-A + cutout [30] | $3.34 \pm 0.06$ | 3.2 | 3150 |
| AmoebaNet-B + cutout [30] | $\mathbf{2.55 \pm 0.05}$ | **2.8** | 3150 |
| Hierarchical Evo [19] | $3.75 \pm 0.12$ | 15.7 | 300 |
| PNAS [18] | $3.41 \pm 0.09$ | 3.2 | 225 |
| DARTS (1st order bi-level) + cutout [20] | $3.00 \pm 0.14$ | 3.3 | 1.5 |
| DARTS (2nd order bi-level) + cutout [20] | $\mathbf{2.76 \pm 0.09}$ | 3.3 | 4 |
| SNAS (single-level) + cutout [37] | $2.85 \pm 0.02$ | **2.8** | 1.5 |
| SMASH [2] | 4.03 | 16 | 1.5 |
| ENAS + cutout [27] | 2.89 | 4.6 | 0.5 |
| BASE (Multi-task Prior) | 3.18 | 3.2 | 8 Meta |
| BASE (Imagenet32 Tuned) | 3.00 | 3.3 | 0.04 Adap / 8 Meta |
| BASE (CIFAR10 Tuned) | **2.83** | **3.1** | 0.05 Adap / 8 Meta |

**Featurization Layers**   To conduct architecture search on these multi-sized, multi-task datasets, the meta-network uses separate initial featurization layers (heads) for each image size. The use of non-shared weights for the initial image featurization both allows the meta-network to learn a better prior as well as enabling the use of different striding in the heads to compensate for the significant difference in image sizes. The Imagenet224 head strides the output to 1/8th of the original input while the 32x32 and 64x64 heads both stride to 1/2th the original input size.

## 5.2   Search Performance

We validated our meta-network by transferring the results of architectures optimized for CIFAR10, SVHN, and Imagenet224 to full-sized networks. Details of how we trained the full networks can be found in Appendix A.1. To derive the full-sized Imagenet architectures, we select a high probability architectures from the posterior distribution of architectures given random 10-class Imagenet224 datasets by averaging the sampled architecture distributions for 8 random datasets. To derive the CIFAR10 and SVHN architectures, we adapted the network on the unseen datasets and selected the architecture with the highest probability of being chosen. The meta-network was trained for 130 epochs. At each epoch, we sampled and trained on a total of 24 tasks, sampling 8 10-class discrimination tasks each from Imagenet32, Imagenet64, and Imagenet224. All experiments were conducted with Nvidia 1080 Ti GPUs.

**Performance on CIFAR10 Dataset**   The result of our Meta Architecture Search on CIFAR10 can be found in Table 1. We compared a few variants of our methods. BASE (Multi-task Prior) is architecture derived from training on the multi-task Imagenet datasets only without further fine-tuning. This model did not have access to any information on the CIFAR10 dataset and is used as a baseline comparison.

The BASE (Imagenet32 Tuned) is the network derived from the multi-task prior fine-tuned on Imagenet32. We chose Imagenet32 since it has the same image dimension as CIFAR10. It does slightly better than the BASE (Multi-task Prior) on CIFAR10. We compare these networks to the BASE (CIFAR10 Tuned), which is the network derived from the meta-network prior fine-tuned on CIFAR10. Not surprisingly, this network performs the best as it has access to both the multi-task prior and the target dataset. One thing to note is that for BASE (Imagenet32 Tuned) and BASE (CIFAR10 Tuned), we only fine-tuned the meta-networks for 0.04 GPU days and 0.05 GPU days respectively. The adaptation time required is significantly less than that required for the initial training of the multi-task prior, as well as the required search time for the rest of the baseline NAS algorithms. With respect to the number of parameters, our models are comparable in size with to the baseline models. Using adaptation from our meta-network prior, we can find high performing models while using significantly less compute.

Table 2: Classification Accuracies on SVHN

| Architecture | Top-1 Test Error | Parameters (M) | Search Time (GPU Days) |
|---|---|---|---|
| WideResnet [38] | **1.30 ± 0.03** | 11.7 | - |
| MetaQNN [1] | 2.24 | **9.8** | 100 |
| DARTS (CIFAR10 Searched) | **2.09** | **3.3** | 4 |
| BASE (Multi-task Prior) | 2.13 | 3.2 | 8 Meta |
| BASE (Imagenet32 Tuned) | 2.07 | 3.3 | 0.04 Adap / 8 Meta |
| BASE (SVHN Tuned) | **2.01** | **3.2** | 0.04 Adap / 8 Meta |

Table 3: Classification Accuracies on Imagenet

| Architecture | Top-1 Err | Top-5 Err | Params (M) | MACs (M) | Search Time (GPU Days) |
|---|---|---|---|---|---|
| NASNet-A [43] | 26.0 | 8.4 | 5.3 | 564 | 1800 |
| NASNet-B [43] | 27.2 | 8.7 | 5.3 | **488** | 1800 |
| NASNet-C [43] | 27.5 | 9.0 | **4.9** | 558 | 1800 |
| AmoebaNet-A [30] | 25.5 | 8.0 | 5.1 | 555 | 3150 |
| AmoebaNet-B [30] | 26.0 | 8.5 | 5.3 | 555 | 3150 |
| AmoebaNet-C [30] | **24.3** | **7.6** | 6.4 | 570 | 3150 |
| PNAS [18] | 25.8 | 8.1 | 5.1 | 588 | 225 |
| DARTS [20] | **26.9** | **9.0** | 4.9 | 595 | 4 |
| SNAS [37] | 27.3 | 9.2 | **4.3** | **522** | 1.5 |
| BASE (Multi-task Prior) | 26.1 | 8.5 | **4.6** | **544** | 8 Meta |
| BASE (Imagenet Tuned) | **25.7** | **8.1** | 4.9 | 559 | 0.04 Adap / 8 Meta |

**Performance on SVHN Dataset** The result of our Meta Architecture Search on SVHN are shown in Table 2. We used the same multi-task prior previously trained on the multi-scale Imagenet datasets and quickly adapted the meta-network to SVHN in less than an hour. We also trained the CIFAR10 specialized architecture found in DARTS [20]. The adapted network architecture achieves the best performance in our experiments and has comparable performance to other work for the model size. This also validates the importance of task-specific specialization since it significantly improved the network performance over both our multi-task prior and Imagenet32 tuned baselines.

**Performance on ImageNet Dataset** The results of our Meta Architecture Search on Imagenet can be found in Table 3. We compare BASE (Multi-task Prior) with Base (Imagenet Tuned), which is the multi-task prior tuned on 224x224 Imagenet. The performance of our Imagenet Tuned model actually exceeds that of existing differential NAS approaches DARTS [20] and SNAS [37] on both top-1 Error and top-5 error. In terms of number of parameters and Multiply Accumulates(MAC), our found models are comparable to state-of-the-art networks. Considering running time, while the multi-task pretraining took 8 GPU days, we only needed 0.04 GPU days to adapt to full sized Imagenet. In Figure 2, we compare our models with other NAS approaches with respect to top-1 error and search time. For fairness, we include the time required to learn the architecture prior, and we still achieve significant accuracy gains for our computational cost.

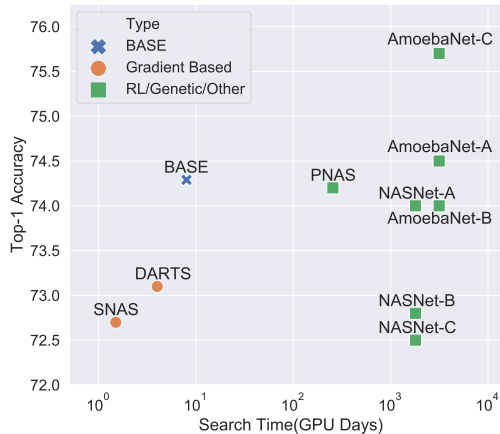

Figure 2: Top-1 Imagenet Accuracy vs Search Time in GPU Days of different NAS methods on Imagenet.

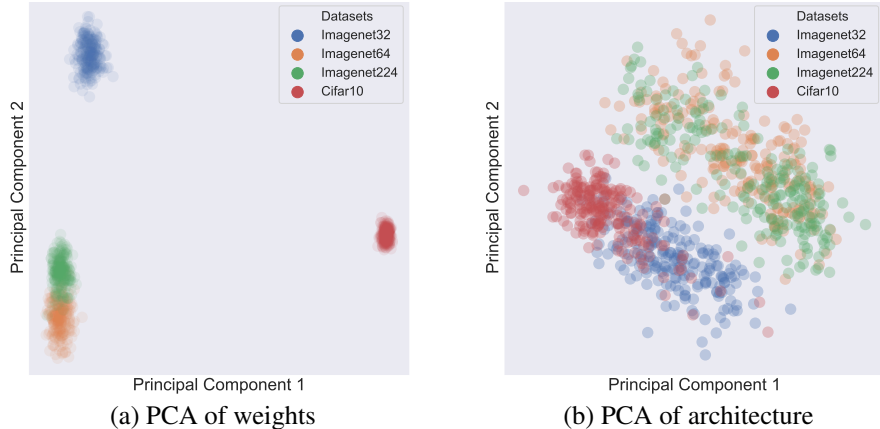

| (a) PCA of weights | (b) PCA of architecture |

Figure 3: Visualization of the PCA for $(\theta, z)$, *i.e.*, weight and architecture, sampled from the posterior distribution of the meta-network.

## 6 Empirical Analysis

In this section, we analyze the task-dependent parameter distributions derived from meta-network adaptation and demonstrate the abilities of the proposed method for fast adaptation as well as architecture search for few-shot learning.

### 6.1 Visualization of Posterior Distributions

Figure 3 shows the PCA visualization of the posterior distributions of the convolutional weights $\psi_D^t$ and architecture parameters $\phi_D^t$. The CIFAR10 optimized distributions were derived by quick adapting the pretrained meta-network for the CIFAR10 dataset while the other distributions were adapted for tasks sampled from the corresponding multi-task datasets. We see that the distribution of weights is more concentrated for CIFAR10 than for other datasets, likely since it corresponds to a single task instead of a task distribution. It also seems that the Imagenet224 and Imagenet64 posterior weight and architecture distributions are close to each other. This is likely due to the fact they are the closest to each other in feature resolution after being strided down by the feature heads to $28 \times 28$ and $32 \times 32$. Considering the visualization of the architecture parameter distributions, it's notable that while the closeness of clusters seems to indicate a similarity between Imagenet32 and CIFAR10, CIFAR10 still has a clearly distinct cluster. This seems to support that even though the meta-network prior was never trained on CIFAR10, an optimized architecture posterior distribution can be quickly derived for CIFAR10.

### 6.2 Fast Adaptations

In this section, we explore the direct transfer of both architecture and convolutional weights from the meta-network by comparing the test accuracy we get on CIFAR10 with meta-networks adapted for six epochs. The results are shown in Figure 4. We compare against the baseline accuracy of the DARTS [20] supernetwork trained from scratch on CIFAR10. Our meta-network adapted normally from a multi-task prior, achieves an accuracy of around 0.75 after only one epoch. We also experimented with freezing the architecture parameters, which greatly degraded the performance. This shows the importance of co-optimizing both the weight and architecture parameters.

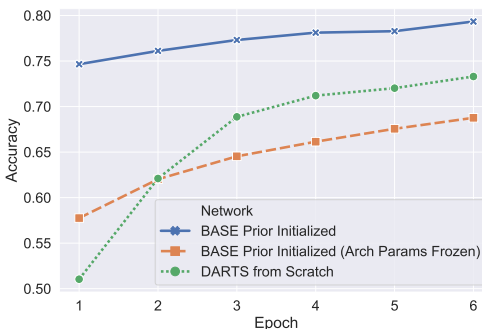

Figure 4: Graph showing the fast adaptation properties of pretrained meta-networks when adapting to CIFAR10 in a few epochs.

Table 4: Comparison of few-shot learning baselines against MAML [8] using the architectures found by our BASE algorithm on few-shot learning on the `Mini-Imagenet` dataset.

| Architecture | 5-shot Test Accuracy | Params (M) | Few-shot Algorithm |
|---|---|---|---|
| MAML [8] | $63.11 \pm 0.92\%$ | 0.1 | MAML |
| REPTILE [26] | $65.99 \pm 0.58\%$ | 0.1 | REPTILE |
| DARTS Architecture | $63.95 \pm 1.1\%$ | 1.6 | MAML |
| BASE (Softmax) | $65.4 \pm 0.74\%$ | 1.2 | MAML |
| **BASE (Gumbel)** | $\mathbf{66.2 \pm 0.7\%}$ | **1.2** | MAML |

## 6.3 Few-Shot Learning

In order to show the generalizability of our algorithm, we used it to conduct an architecture search over the few-shot learning problem. Since few-shot learning targets adapting in very few samples, we can avoid using the Finite Difference approximation and directly use the optimization-embedding technique in these experiments. These experiments were run on a commonly used benchmark for few-shot learning, the `Mini-Imagenet` dataset as proposed in [33], specifically on the 5-way classification 5-shot learning problem. The full-sized network is trained on the few-shot learning problem using second-order MAML [8]. Search and full training were run twice for each method. A variation of our algorithm was also run using a simple softmax approximation of the Categorical distribution as proposed in [20] to test the effect of the Gumbel-Softmax architecture parameterization. The full results are shown in Table 4, our searched architectures achieved significantly better average testing accuracies than our baselines on five-shot learning on the `Mini-Imagenet` dataset in the same architecture space. The `CIFAR10` optimized DARTS architecture also achieved results that were significantly better than that found in the original MAML baseline [8] showing some transferability between `CIFAR10` and meta-learning on `Mini-Imagenet`. That architecture, however, also had considerably more parameters than our found architectures and trained significantly slower. The Gumbel-Softmax meta-network parameterization also found better architectures than the simple softmax parameterization.

## 7 Conclusion

In this work, we present a Bayesian Meta-Architecture search (BASE) algorithm that can learn the optimal neural network architectures for an entire task distribution simultaneously. The algorithm derived from a novel Bayesian view of architecture search utilizes the optimization embedding technique [6] to automatically incorporated the task information into the parameterization of the posterior. We demonstrate the algorithm by training a meta-network simultaneous on a distribution of $2.634 \times 10^{23}$ tasks derived from `Imagenet` and achieve state-of-the-art results given our search time on both `CIFAR10`, `SVHN`, and `Imagenet` with quick adapted task-specific architectures. This work paves the way for future extensions with Meta Architecture Search such as direct fast-adaption to derive both optimal task-specific architectures and optimal weights and demonstrates the great efficiency gains possible by conducting architecture search over task distributions.

## Acknowledgments

We would like to thank the anonymous reviewers for their comments and suggestions. Part of this work was done while Bo Dai and Albert Shaw were at Georgia Tech. Le Song was supported in part by NSF grants CDS&E-1900017 D3SC, CCF-1836936 FMitF, IIS-1841351, SaTC-1704701, and CAREER IIS-1350983.

## Footnotes

*Corresponding author: `ashaw596@gatech.edu`

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
