[Supplementary Material]

# Appendix

## A  Architecture Space Details

For comparability in architectures, the particular search space used is very similar to that used in [20] and includes the same operation space: $3 \times 3$, $5 \times 5$, $7 \times 7$ depth-wise separable convolutions, $3 \times 3$ and $5 \times 5$ dilated depth-wise separable convolutions, $3 \times 3$ max pooling, $3 \times 3$ average pooling, a $1 \times 7$ followed by a $7 \times 1$ convolution, skip connections, and no connection. In our search, each cell is made up of a total of six nodes with 2 input nodes. The input to each cell is the output from the previous 2 cells. The output for each cell is the concatenated output from all 4 non-input nodes in the cell. Following the same methods as [20, 43], non-dilated depth-wise separable convolutions were applied twice, all depth-wise separable convolutions did not have batch-norms between the grouped and 1x1 convolutions, convolutions had RELUs and batch-norms applied in ReLU-Conv-BN order, and all operations were padded as necessary to preserve spatial resolution as to only be reduced by the reducing layers whose first operations were applied with a stride of 2.

### A.1  `CIFAR10` and `Imagenet` Training Details

`CIFAR10`   The architecture is transferred to a network with 20 cells following the motif shown in Appendix A.2. The network was trained for 600 epochs with cutout augmentation. We used a batch size 96. We follow the same training strategy as [20] with cutout, and drop-path probability of 0.2, and auxiliary towers with weight 0.4.

`SVHN`   The architecture is transferred to a network with 20 cells following the motif shown in Appendix A.2. The network was trained for 160 epochs. We used a batch size 96, a drop-path probability of 0.2, and auxiliary towers with weight 0.4. The networks were trained for 160 epochs with cutout augmentation.

`ImageNet`   The architecture is transferred to a network with 14 cells following the motif shown in Appendix A.2. We train and evaluate in the mobile setting with input images of size 224x224. We train with a batch size of 256 for 375 epochs. We use the SGDR[24] learning rate schedule with $T_0 = 25$ and $T_m ult = 2$. We optimize with the SGD with a initial lr of 0.1 decayed by a factor of 0.97 each epoch. We use a weight decay of $3e^{-5}$. For the remaing parameters we follow the same training strategy as [43].

## A.2    Motifs for Single-Task Scalable Architectures

Motif for the Search Network

Motif for `CIFAR10` Full Network.

Motif for `ImageNet` Full Network.

These are the network motifs used in the experiments for search over single-task networks. Our search space has two unique cell architectures, "Normal Conv" and "Reduction" Cells.

## A.3    Sample `ImageNet` Adapted Cell Designs

Cell Design for normal cell

Cell Design for reduction cell

# B  Few Shot Learning

## B.1  Motifs for Scalable Architectures

Motif for Search Network                    Motif for Full Network.

These are the network motifs used in the experiments for search over few-shot learning. Our search space has two unique cell architectures, "Normal" and "Reduction" Cells. The Meta Architecture Search was run with the "Search Network", and then for evaluation, the architectures were transferred to the full network.

## B.2  High Level Diagrams of the Meta Architecture Search method.

(a) Meta Architecture Search          (b) One-Shot Architecture Adaptation

## B.3  Diagram of Cell space concept

The architecture parameters $\phi_{Normal}$ are shared between all architecture "normal cells" and describe the architecture distribution within in the cells. $\phi_{Reduce}$ are shared between all reduce cells. All $\psi$ weight parameters are not unique to each layer.

## B.4 Few-shot Training Details

In our experiments on the `Mini-Imagenet` dataset, only the 64 training classes were used during training. The 12 validation classes were ignored, and evaluation was conducted on the 24 testing classes. Search was run for 10000 iterations. For each iteration, the meta-network was updated with the combined gradients from $T = 2$ randomly sampled tasks. For each task $N = 4$ steps of inner optimization were run. For the full training, all network architectures were trained with the same setting on the 5-shot learning problem using the second-order MAML algorithm [8]. The full training was run for 30000 iterations. Similarly, for each iteration, the network was again updated with the combined gradients from 2 randomly sampled tasks, but each task was optimized with 5 steps of inner optimization for second-order MAML.

## B.5 Sample Top Found Cell Architectures from few-shot BASE search

Cell Design for sample normal cell

Cell Design for sample reduction cell