[Reviews · NeurIPS 2019]

Reviewer 1



Originality: The work is original in its modelling of meta architecture search. The approach used builds upon existing methodologies in variational inference (coupled VB) and neural architecture search (bayesian formalism of DARTS search space) (Perhaps also similar to [1] in the use of Gumbel Softmax trick) Quality: The work is well presented. Relevant baselines have been cited in the comparative study. An empirical analysis to understand different aspects of the proposed model is also provided. Clarity: This work clearly motivates the problem statement, provides a solution and supports it with an empirical study to understand the learnt posterior distribution of different tasks, the effect of prior and the flexibility of model in few-shot learning setting. Significance: As NAS methods are often expensive, it is natural to look for meta modelling approaches that can help share certain aspects of NAS across tasks. This work is a useful step in this direction. Weaknesses: The tasks considered comprise of different resolutions of the same dataset. It is well known that models learns on ImageNet transfer to different resolutions. Evaluating the approach on a diverse set of tasks would be useful. General comments A latex typo on Page 5 \texttt{CIFAR-10} [1] Bichen Wu, Xiaoliang Dai, Peizhao Zhang, Yanghan Wang, Fei Sun, Yiming Wu, Yuandong Tian, Peter Vajda, Yangqing Jia, Kurt Keutzer: FBNet: Hardware-Aware Efficient ConvNet Design via Differentiable Neural Architecture Search. CoRRabs/1812.03443 (2018)

Reviewer 2



The authors propose Bayesian Meta Architecture Search (BASE), a method for meta learning neural network architectures and their weights across tasks. The paper frames this problem as an Bayesian inference problem and employs Gumbel-Softmax, reparametrization and optimization embedding, a variation inference method, to optimize a distribution over neural network architectures and their weights across different tasks. Originality: Meta learning neural network architectures is a very natural next step for NAS research, which as not been done so far (at least I’m not aware of any work). It is not only very natural but also very important as it allows to make NAS more scalable and of more practical relevance. The Bayesian view, however, is not really novel, but rather an obvious extension of [1]. [1] should have been discussed more thoroughly and the authors should point out differences (if any more). In general, the related work section is very short and does not provide a proper summary of the current state of the art in this field of research Quality: BASE is well motivated and derived. However, it is unclear to me which role the optimization embedding actually plays. Is it actually used? Which part of the equations and pseudo code are not already explainable/covered by ELBO+Gumbel-Softmax + reparametrization? The empirical results seem convincing (in terms of search time) at first glance, but I have the following concerns: (i) while the authors claim that BASE significantly speeds up NAS by meta learning (which I tend to agree with), they do not properly comment on how expensive the training of the full model actually is. It seems that this final training is actually more expensive than the search process and should therefore be considered. Are the meta-learned weights actually used for the full sized model? If so, how? (ii) Missing ablation studies/ unfair comparisons. A proper ablation study on the meta learning of the architecture is missing. E.g., in section 6.2., one could run BASE without architecture adaptation (i.e., only meta learning the weights) during the meta learning phase. How would DARTS perform if one would simply pre-train it on the meta-train tasks rather than running DARTS from scratch? This would be a more fair comparison. In 6.3., to the best of my knowledge, MAML and REPTILE use the same architecture, so why are they both mentioned in the table and with different results? Also, this architectures has approx.. 100k params, which is not stated in the table. REPTILE also seems to be on-par with BASE(Gumbel) while likely requiring significantly less compute,. (iii) The error numbers should be taken with a grain of salt as it more and more seems that the cell-based search space in combination with scaling up found models is less interesting and apropriate than one would think [2,3]. In terms of error rates, I do not see significant improvements over prior work (which is of course not the focus of this work). However, the authors tend to overclaim their results (e.g., in the abstract: “This result beats the state-of-the-art methods such as DARTS and SNAS significantly in terms of both performance and computational costs”). * Clarity: The paper is well structured, even though it would be better to have the related work section in the beginning. As mentioned before, the related work section is rather short. It would be helpful for the reader to have some information on optimization embedding as well. The paper contains various typos and bad grammar, which makes it hard to read the paper. The paper is partially overloaded with notation and sometimes the notation is not explained and /or not re-used and / or not consistent with the figures. E.g., in Figure 1, one can only assume that “NN” means neural network. What are \phi, \psi? It is unclear at this point but only mentioned later in the paper. In the right part of Figure 1, what is 1,2,..J? What is M? Is notation is not used in the text (as far as I can see). Equation (1), why introduce A? Equation (2), what are f and l? Figure 2 is barely readable. * Significance: The authors address an important problem, namely extending NAS methods to the meta learning setting, so that one does not need to re-run NAS from scratch for a new task. However, it is not clear if this paper achieves due to the concerns on the empirical evaluation mentioned above. The methodological novelty right now is vague to me as (i) there seems to be large overlap with [1] and (ii) the role and impact of optimization embedding is not fully clear. In summary, the authors present an interesting approach on using architecture search in a meta learning setting. Empirical results need to be taken with a grain of salt. The presentation of the paper is borderline. [1] SNAS: stochastic neural architecture search, Xie et al., ICLR 2019 [2] Evaluating the Search Phase of Neural Architecture Search, Sciuto et al. [3] Random Search and Reproducibility for Neural Architecture Search, Li et al. ------------------------------------------- Update after author feedback --------------------------------- Thank you for commenting on my review. I acknowledge that this work does indeed give a better Bayesian picture on NAS and extends it to the meta learning setting, which is an improvement over SNAS. However, I still have one major concern, that has not been addressed in the rebuttal: the authors only compare to NAS methods ran from scratch. There is no baseline that employs the meta train set. E.g., one important baseline that should have been compared to: pre-running some NAS method, e.g. DARTS, on the meta train set + adaptation on the meta test set (for the same compute time as BASE). Therefore, the benefit of the proposed method is still unclear to me and I decided to not increase my rating.

Reviewer 3



The idea seems to be novel to me. I specifically, like the skip-connection part which allows one to compose network with different functioning layers. The only limitation is that the number of such elements are hard-limited to K and it is not permutation-invariant. However, I think the paper still stands without them. The experimental results have been presented elaborately. It is also very interesting to look at the PCA results of the weights and the architecture and how the different datasets are organised with respect to others. cifar10 and imagenet32 are close in the architecture space, but different in the weight space - which makes sense as the image sizes are close. I am not a deep learning person, so I am unable to see the limitation from the real application point of view. But looking from the meta-problem and the Bayesian methodology, I would love to accept this paper. Minor comments are: 1. The paper needs a good proof reading to fix many type-setting issues e.g. extra 'a' in line 19, extra 'then' in line 53, part of line 66 can be writte better as "...network as a composition of L layers of cells...", extra textit in line 198. 2. The authors also have not spent any effort in giving proper references: sometimes conference names are in long format, somethimes they are short. Even it has duplicate references, [24] and [25]. Please fix them in the later version.

[Author Response · NeurIPS 2019]

We would like to thank all the reviewers for careful reading our paper, and we sincerely appreciate the constructive comments. We will individually address the questions raised by each reviewer.

**Reviewer 1** **Transferability to diverse datasets.** We conducted additional experiments on the `SVHN` dataset shown in the Table to further demonstrate our algorithm. We used the same Meta network prior previously trained on the multi-scale `Imagenet` datasets and quickly adapted the meta network to `SVHN` for two epochs in less than an hour. The networks were trained for 160 epochs with cutout augmentation. We also trained the `CIFAR10` specialized network

| Architecture | Top-1 Err | Params(M) |
|---|---|---|
| WideResnet [2] | $1.30 \pm 0.03$ | 11.7 |
| MetaQNN [3] | 2.24 | 9.81 |
| DARTS (`CIFAR10` Searched) | 2.09 | 3.3 |
| BASE (Multitask Prior) | 2.13 | 3.22 |
| BASE (`Imagenet32` Tuned) | 2.07 | 3.29 |
| BASE (`SVHN` Tuned) | **2.01** | 3.22 |

from DARTS. The adapted network achieves the best performance in our experiments and has comparable performance to other work for the model size. It is difficult to compare to [1] since they don't explicitly state their final performance on the `SVHN` dataset outside of small graphs and search smaller networks. Roughly, based on the figures, it appears to achieve about 97% accuracy in a 2 GPU hours and around 97.3% accuracy in 20 GPU hours.

**Reviewer 2** **Novelty of the Bayesian View.** Our Bayesian view of architecture search is fundamentally different from SNAS [4] since our approach has the ability to naturally extend to meta architecture search while SNAS cannot. Specifically, we can transfer prior information among tasks which is critical for meta architecture search. SNAS's formulation which is motivated from the policy gradient in reinforcement learning, has no prior structure and can only be applied to a single task. Compared with SNAS, the networks found by BASE achieved comparable performance on *CIFAR10* and higher accuracy on *Imagenet* with meta architecture search.

**Optimization Embedding.** Optimization embedding is used for parameterization of the posterior distribution of the convolutional weights and architecture parameters in Eq. (11), $\phi_{\mathcal{D}}^t$ and $\psi_{\mathcal{D}}^t$. In the few-shot learning experiments, we fully unroll the optimization embedding while in the other experiments, we use the finite difference approximation corresponding to line 10 of algorithm 1. Compared to the vanilla independent posterior parametrization with an arbitrary neural network, optimization embedding allows us to model the dependencies between $D$ and $W$ *explicitly* which allows to better adaptation to new datasets.

**Training Time.** As with most work in the field of architecture search, our method focuses on architecture search over model training. However, with our low search times, we do recognize the increasing importance of training time, and we will add a discussion to our final paper. Our Bayesian formulation does naturally allow transfer of weights from the meta-network if we don't transfer to larger networks which could potentially eliminate the additional training time.

**Transferability of Meta-Weights.** We did not transfer the posterior of weights to the final networks in our experiments. Figure 4, however, does demonstrate the fast adaptation of both weights and architecture where our meta-network achieves 75% accuracy on the unseen `CIFAR10` in one epoch based on the posterior of both weights and architecture.

**Clarification about few-shot learning results.** Indeed, MAML and REPTILE use the same model architectures. However, they are fundamentally different algorithms for Meta-learning: MAML uses the full gradient and REPTILE uses the finite difference approximation, leading to different performances.

To be clear, DARTS, BASE (Softmax), and BASE (Gumbel) are the algorithms used to generate network architectures. They are *not* few-shot classification algorithms and in Table. 3, the actual few-shot learning with which the models are trained is conducted using MAML. Therefore, it only makes sense to directly compare against the MAML baseline. The REPTILE baseline is included only for completeness. It is true that the models used in MAML were much smaller, and we will update the paper. However, we chose to use our larger search space so we could directly compare to the the `CIFAR10` optimized network architecture from DARTS. This network, though much larger, significantly beat the original MAML paper results with the same algorithm. BASE was able to find networks in the same search space which achieved even better results. We will make this point clearer in our final paper.

**Error Rates.** It is definitely possible that the cell space and scaling may complicate the results, but this is still a common methodology and shouldn't invalidate our work. I believe the particular part you quoted from our paper was in reference to our found Imagenet-optimized architecture which achieved fairly high performance, but we will revise those sections to clarify our claims.

**Clarity of Figures.** We will be expanding the related work, clarifying the notation, and fixing those typos. To address the issue in Figure 1, the $M$ in that figure should be $L$ following the notation in the rest of the paper which corresponds to the number of choices of operations for each layer type.

**Reviewer 3** Thank you for your appreciation and suggestions for paper refinement. We will polish the the final version of our paper following your comments.

[1] Wistuba et al. Inductive Transfer for Neural Architecture Optimization. *arXiv:1903.03536*, 2019.
[2] DeVries et al. Improved regularization of convolutional neural networks with cutout. *arXiv:1708.04552*, 2017.
[3] Baker et al. Designing neural network architectures using reinforcement learning. *ICLR* 2017.
[4] Xie et al. SNAS: stochastic neural architecture search. *ICLR* 2019.


[Meta-Review · NeurIPS 2019]

This paper shows how to apply transfer learning from related tasks to NAS. It proposes a variational Bayesian formulation, which is related to earlier work, but the transfer learning approach is novel, and is clearly useful, given that NAS people can assemble a sufficient number of tasks over time. As detailed in the reviews, the experimental validation could be improved.